
# Deriving clear-sky longwave spectral flux solely from hyperspectral radiance: a case study with AIRS observations

Xiuhong Chen*, Xianglei Huang

Department of Climate and Space Sciences and Engineering, the University of

Michigan at Ann Arbor.

* Corresponding author: Dr. Xiuhong Chen, Department of Climate and Space Sciences and

Engineering, the University of Michigan, Ann Arbor, MI 48109-2143, USA. (Tel) 734-647-7687.

xiuchen@umich.edu



1          Abstract

2          Previous studies have shown that longwave (LW) spectral fluxes have unique merit in

climate studies. Using Atmospheric Infrared Sounder (AIRS) radiances as a case study, this study
presents an algorithm to derive the entire LW clear-sky spectral fluxes solely from hyperspectral
observations. No other auxiliary observations are needed in the algorithm. A clear-sky scene is
identified using a three-step detection method. The identified clear-sky scenes are then
categorized into different sub-scene types using AIRS radiances at six selected channels. A
previously established algorithm is then used to invert AIRS radiances to spectral fluxes over the
entire LW spectrum at 10 cm$^{-1}$ spectral interval. Accuracy of the algorithms is evaluated against
collocated Clouds and the Earth's Radiant Energy System (CERES) observations. For nadir-view
observations, the mean difference between outgoing longwave radiation (OLR) derived by this
algorithm and the collocated CERES OLR is 1.52 Wm$^{-2}$ with a standard deviation of 2.46 Wm$^{-2}$.
When the algorithm is extended for viewing zenith angle up to 45°, the performance is
comparable to that for nadir-view results.
Key words: longwave spectral flux; OLR; clear-sky detection; sub-scene type classification;
hyperspectral observations; AIRS



## 1. Introduction

Broadband outgoing longwave radiation (OLR) obtained by Earth's Radiation Energy Balance (ERBE; Barkstrom 1984) and Clouds and the Earth's Radiant Energy System (CERES; Wielicki et al., 1996) has been extensively used in climate studies for three decades. The physical quantity directly measured by the ERBE or CERES instruments is actually a convolution between broadband upwelling radiance at a given viewing zenith angle and the spectral response function (SRF) of the broadband radiometer on the EREB or CERES. Then the broadband upwelling radiance is inferred through deconvolution of the measurement and, consequently, it is converted to broadband flux (e.g. Loeb et al., 2005; Kato and Loeb, 2005). In order to reliably derive the broadband flux, a variety of auxiliary information needs to be used to define the scene type for each instrument footprint. Such auxiliary information includes, but is not limited to, surface temperature, lapse rate, precipitable water, and cloud macroscopic properties (e.g. cloud fraction, cloud emissivity). For the case of CERES, such auxiliary information is obtained from other satellite measurements such as MODIS and SSM/I as well as operational analysis (Loeb et al., 2005).

The integrand of broadband OLR, the spectral flux, is not available from the broadband flux measurements such as ERBE or CERES because of the nature of broadband radiometer in these measurements. However, the spectral flux can provide critically valuable information for the climate model diagnostics. Especially, comparing modeled and observed spectral flux can expose compensating biases in the simulated radiation budget by the climate model that otherwise cannot be exposed from broadband flux diagnostics alone (Huang et al., 2006; Huang



et al., 2013; Huang et al., 2014). Similarly, spectral cloud radiative forcing can also help expose

compensating biases from different bands (Huang et al., 2013; Huang et al., 2014).

Currently there are several operational hyperspectral sounders in space that measure

spectral radiances in thousands of IR channels, for example, Atmospheric Infrared Sounder

(AIRS; Aumann et al., 2003) since 2002, Infrared Atmospheric Sounding Interferometer (IASI;

Hilton et al., 2012) since 2006, and Cross-track Infrared Sounder (CrIS; Han et al., 2013; Strow et

al., 2013) since 2011. Each of these sounders can acquire several millions of spectra per day. A

series of studies published in recent years (Huang et al., 2008, 2010, 2014; Chen et al., 2013)

have established algorithms to estimate observation-based spectral flux from the AIRS

radiances using the scene type information from collocated CERES footprints. Specifically,

spectral angular distribution models (ADMs) for each AIRS channel have been constructed for

the scene types defined for the CERES SSF (Single Satellite Footprint) data set and then applied

to AIRS radiances to derive spectral flux at each AIRS channel. The spectral ADMs are trained

from synthetic AIRS radiances and the meteorological fields from the ECMWF ERA-Interim

reanalysis (Dee et al., 2011) that are used to generate the synthetic radiances. A principal

component-based multivariate linear regression scheme is then used to estimate spectral flux

over the spectral bands not covered by the AIRS instrument. The end product is spectral flux at

10 cm$^{-1}$ interval over the entire LW spectrum. The spectral flux derived from this method has

been extensively compared with collocated CERES OLR and the agreement is robustly

consistent across different scene types and over different spatial and time scales, from

footprint level to gridded data, from monthly means to annual means and interannual

variations (Huang et al., 2008, 2010, 2014; Chen et al., 2013).



The aforementioned series of studies took a shortcut by relying on the scene type
information from collocated CERES dataset. The other hyper-spectral sounders such as CrIS and
IASI also fly with imagers such as VIIRS and AVHRR, respectively. These imagers provide
information needed for scene type classification. However, to apply information from these
imagers, the near-simultaneous observations as well as the collocation strategy are required to
overcome the differences in observational area and time period (Huang et al., 2008; O'Carroll et
al., 2012; Wang et al., 2013). The rich information contained in the hyperspectral radiances
naturally leads to a hypothesis that all information needed for defining scene types might be
already contained in the spectral radiances. Therefore, a scientifically intriguing question to ask
is: can we directly estimate spectral flux from such observations of hyperspectral radiances
without relying on auxiliary observations and thus avoid the trouble of collocation strategy and
reduction of samples? To follow this line of thinking, this study explores ways of defining scene
types and sub-scene types solely from hyperspectral measurements such as AIRS radiances, and
then evaluates the spectral flux derived in this manner. As a first step, we  focus on clear-sky
scene types in this study. This effort aims to estimate longwave spectral flux and broadband
OLR directly from AIRS Level-1 calibrated radiances over each individual single footprint. This
approach is different from other studies such as Dessler et al. (2008), Moy et al. (2010) and
Susskind et al. (2012), which fed temperature and humidity fields from AIRS Level-2 retrievals
(defined for 3-by-3 AIRS footprints) or even Level-3 monthly gridded data set into a radiative
transfer model to compute the clear-sky OLR. Huang et al. (2008, 2010, 2014) and Chen et al.
(2013) have demonstrated that such direct estimate of spectral flux from AIRS radiances is
feasible and the estimated OLR highly agree with the collocated CERES OLR. Furthermore, the



merit of the spectral flux in testing climate models also warrants a feasibility study of deriving
spectral flux (preferably over the entire longwave spectrum) from the hyperspectral satellite
observations. All these facts have motivated the study presented in this article.

The rest of this paper is organized as follows. Section 2 describes the dataset and forward

model used in this study. Clear-sky detection, sub-scene type classification, and the derivation
of spectral flux for the case of nadir-view observations are described in Section 3. Section 4
validates the overall algorithm mentioned in Section 3. Section 5 discusses performances of the
algorithm in other viewing zenith angles within ±45°. Conclusions and further discussion are
then presented in Section 6.
**2. Data sets and forward model**

The data sets and forward model used in this study are identical to those used in (Huang

et al., 2008; Chen et al., 2013). Below is a brief depiction of the relevant features of data and
forward model.

AIRS is an infrared grating array spectrometer aboard NASA's Aqua satellite launched in

2002 (Aumann et al., 2003). It measures radiances across three bands, 3.74-4.61 μm, 6.20-8.22
μm and 8.8-15.4 μm, with a spectral resolving power ($\lambda/\Delta\lambda$) of ~1200, which converts to
approximate full width at half max (FWHM) resolutions of ~0.5 cm$^{-1}$ at 650 cm$^{-1}$ and ~2.0 cm$^{-1}$ at
2500 cm$^{-1}$. It scans from -49° to 49° across the track with 13.5-km ground footprints at the nadir
view. This study uses AIRS level-1b calibrated radiances in the entire year of 2004.

For the purpose of validation, broadband OLR and sub-scene type information from the

Aqua-CERES SSF Edition 3 are used. The strategy to collocate CERES and AIRS observations at



the footprint level is the same as described in Huang et al. (2008). The CERES SSF algorithm
employs a MODIS-imagery based algorithm to detect clear-sky footprint (Geier et al., 2003). The
total precipitable water (TPW) in the CERES SSF data set is retrieved from the Special Sensor
Microwave Imager (SSM/I; Goodberlet et al., 1990). Its lapse rate ($\Delta T$) is derived from the GEOS
Data Assimilation System (DAO, 1996). Surface skin temperatures ($T_s$) are estimated from
MODIS clear-sky 11-$\mu$m radiance (Minnis et al., 2004). The CERES SSF algorithm uses $\Delta T$, $T_s$, and
TPW to define sub-scene types of clear-sky observations. Thus, the OLR can be inverted using
appropriated broadband ADM and measured broadband radiances (Loeb et al., 2005; Kato and
Loeb, 2005). Uncertainty of inverted CERES OLR is about 1% (Loeb et al., 2007).
The European Center for Medium range Weather Forecasting (ECMWF) ERA-Interim
reanalysis (Dee et al., 2011) is used in this study as well. It has a spatial resolution of 1.5°
latitude by 1.5° longitude and 37 vertical levels up to 1hPa. Similar to Huang et al. (2008) and
Chen et al. (2013), the forward radiative transfer model used here is the MODerate resolution
atmospheric TRANsmission code (MODTRAN, version 5; Anderson et al., 2007). MODTRAN is
used to compute synthetic AIRS radiances and outgoing spectral fluxes at the top of
atmosphere (TOA). MODTRAN 5 offers a spectral resolution as fine as 0.1 cm$^{-1}$ (higher than AIRS
spectral resolution). Compared with AIRS observations, MODTRAN 5 replicates AIRS brightness
temperatures over 650-1600 cm$^{-1}$ with mean difference of ~0.2 K (the AIRS noise equivalent
delta temperature NEDT being 0.51 K over this band) (Anderson et al., 2007). AIRS radiances are
generated by convoluting MODTRAN output and tabulated spectral response functions of AIRS
channels (Strow et al., 2006). The TOA spectral fluxes are computed using a three-point
Gaussian quadrature (Clough and Iacono, 1995).



### 3. Algorithm for estimating clear-sky LW spectral flux: the case of nadir view


The algorithm for estimating clear-sky LW spectral flux from nadir-view AIRS spectral
radiances consists of three steps. The first step is to use radiance alone to decide whether an
AIRS spectrum can be considered as a clear-sky spectrum or not (usually referred as clear-sky
detection). The second step is to classify the sub-scene type of a clear-sky spectrum using
radiance information alone. The third step is to invert the AIRS radiances to spectral flux over
the entire LW spectrum.
**3.1. Clear-sky detection**
**3.1.1. Algorithm design**
Detecting clear-sky scenes from IR radiance alone is usually done by applying a
sequence of tests (Amato et al., 2014; and references therein). We use three tests in sequence
for this purpose. Test 1 is a spatial inhomogeneous test commonly referred as the "Golden
Arches" test proposed first by Coakley and Bretherton (1983). For a given AIRS footprint and
four adjacent AIRS footprints, if the standard deviation of brightness temperatures at a window
channel 963.8 cm$^{-1}$ (hereafter denoted as $BT_{963.8}$) is smaller than a predetermined threshold
value C1, the footprint passes Test 1. For the footprint that passes the "Golden Arches" test,
Test 2 is a bi-spectral test, namely the brightness temperature difference between two narrow
bands—one being 8 μm band ($BT_8$, 8.17-8.92 μm) and the other being 11 μm band ($BT_{11}$, 10.06-
11.25 μm). Test 2 utilizes the spectrally dependent feature to distinguish cloudy spectrum and
clear-sky spectrum, because the 11 μm band is sensitive to water clouds and ice clouds, while
the 8 μm band has weak water vapor absorption lines, and the $BT_8$-$BT_{11}$ difference has been




widely used for this purpose (e.g. Ackerman and Strabala, 1994). If the $BT_8$-$BT_{11}$ of an AIRS
spectrum is less than a pre-determined value C2, the spectrum passes Test 2. Test 3 is a
threshold test to compare the $BT_{963.8}$ with the surface temperature at the ground footprint
interpolated from 6-hourly ERA-interim reanalysis, termed as $Ts_{ERA}$-$BT_{963.8}$. $BT_{963.8}$ is used as a
surrogate of surface temperature in Chen and Huang (2014) because this channel has little
atmospheric absorption in the case of clear sky. If $Ts_{ERA}$-$BT_{963.8}$ of an AIRS spectrum is smaller
than a pre-determined value C3, the spectrum passes Test 3. Only when a spectrum passes all
three tests, do we deem it to be a clear-sky spectrum. Note that, though ERA-interim reanalysis
is used in this study, in future operational applications the reanalysis surface temperature can
be replaced by the surface temperature from operational analysis.

We used four months of collocated AIRS and CERES nadir-view observations in 2006 to

empirically determine the threshold values used in the three tests (i.e., C1, C2, and C3). In
another words, we use the clear-sky footprint identified by CERES as the "ground truth" and
decide the threshold values based on collocated AIRS observations accordingly. The four
months used for this purpose are January, April, July, and October of 2006. A total of ~1.56
millions of collocated observations are available for this training purpose. We first categorize
the observations into four groups: daytime ocean, nighttime ocean, daytime land, and
nighttime land. Then for each group, the threshold value is defined as the value suitable for
describing 95% of qualified observations. An example of how to decide C1 is given in Figure 1.
Each panel plots the histogram of the standard deviation based on the $BT_{963.8}$ of the clear-sky
AIRS footprint and four adjacent AIRS footprints. Only 5% of clear-sky observations in each
panel have a standard deviation larger than the value denoted by the dash-dot line, which is



then assigned as the value of C1. The value of C2 is decided in a similar manner. Water vapor
continuum absorption is important for the AIRS channel at 963.8 cm$^{-1}$. Such absorption is
dominated by humidity in the planetary boundary layer, which is highly correlated with surface
temperature. Therefore, we divide observations further into different subgroups based on the
value of $Ts_{ERA}$ and the value of C3 is determined for each subgroup accordingly. Table 1
summarizes the threshold values for C1, C2, and C3 derived in this manner.
**3.1.2. Performance of the clear-sky test algorithm**

We assess the performance of the clear-sky test algorithm using collocated CERES and

AIRS nadir-view observations in the entire year of 2004 (4.48 millions of observations in total).
The performance is summarized in Table 2. The false negative (FN) rate refers to the percentage
of collocated CERES clear-sky observations that have been classified as cloudy-sky observations
by our algorithm. The false positive (FP) rate refers to the percentage of collocated CERES
cloudy-sky observations that have been classified as clear-sky observations by our algorithm.
The overall accuracy rate refers to the percentage of cases in which our algorithm can correctly
classify the footprints. It can be seen that, although using three tests together increases the
rate of false negative, such an approach is also effective in reducing the false positive rate.
Given that the number of cloudy-sky observations is ~9-10 times more than that of clear-sky
observations, using three tests together can achieve a better accuracy than using one of the
tests alone. As far as the FN and FP rates are concerned, this algorithm is comparable to other
clear-sky detection algorithms that are based on IR spectral radiances alone (e.g. Table 4 in
Amato et al., 2014).



### 3.2. Sub-scene type classification


The second step in the overall algorithm is to classify the sub-scene types of clear-sky
observations identified by the algorithm described in Section 3.1. The sub-scene types adopted
here are largely similar to the discrete intervals defined by Table 3 in Loeb et al. (2005), which
depend on the total precipitable water (TPW), surface temperature ($T_s$), and lapse rate (ΔT)
defined as temperature difference between the surface and 300 hPa above it. Similar to Chen
and Huang (2014), here $BT_{963.8}$ is used as a surrogate of surface temperature. ΔT is inferred
from brightness temperature differences of two AIRS channels: 963.8 and 748.6 cm$^{-1}$ (hereafter
denoted as $ΔBT_{963.8 - 748.6}$). A quick estimate of TPW is obtained by a look-up-table approach
proposed by Chen and Huang (2014), which makes use of double difference of two pairs of AIRS
channels as well as $BT_{963.8}$ and $ΔBT_{963.8 - 748.6}$ to construct the look-up-table. Table 3 lists the
accuracy of this algorithm based on the collocated AIRS and CERES observations in 2004 and
the comparison with the auxiliary information of TPW, $T_s$, and ΔT in the CERES SSF dataset. It
can be seen that, though this estimate method is solely based on AIRS radiances, the accuracy is
80% or even higher.

### 3.3. Estimate of fluxes from radiance measurements


The last component in our algorithm is to invert spectral fluxes from the AIRS radiances.
Huang et al. (2008) adopted the same sub-scene type classification as in Loeb et al. (2005) for
inverting AIRS radiance to spectral flux. Therefore, the algorithm in Huang et al. (2008) can be
used here without further modification. Specifically, the spectral radiance ($I_{AIRS}(\theta)$ at each
viewing zenith angle $\theta$) is first converted to spectral flux ($F_{AIRS}$) over each AIRS channel using a



pre-calculated spectral ADM ($R_{AIRS}(\theta)$) for each sub-scene type, $F_{AIRS} = \pi \cdot I_{AIRS}(\theta) / R_{AIRS}(\theta)$.
Then a principle component-based multivariate prediction scheme is used to estimate spectral
fluxes over the spectral portion not covered by the AIRS instrument. The performance of this
radiance-to-flux algorithm and its characteristics has been documented in detail in Huang et al.
(2008) and Chen et al. (2013).

**4. Validation of the overall algorithm**

This section focuses on validation of the overall algorithm in terms of its performance in
estimating the spectral flux over the entire longwave spectrum. The following parts (1)-(3)
examine the performance of the scene type classification algorithm, and part (4) examines the
overall performance of the clear-sky detection and the scene type classification algorithms.
(1) We feed 6-hourly ERA-interim reanalysis data to the forward model to simulate
clear-sky AIRS radiances and apply our algorithm to estimate the spectral flux (hereafter $F_{AIRS-}$
$_{only}$). We then compare this spectral flux with clear-sky spectral fluxes directly computed using
the ECMWF ERA-Interim reanalysis with the same forward model (hereafter $F_{ERA}$). This is an
idealized test because the forward modeling is assumed to be a surrogate of reality. Specifically,
6-hourly ERA-interim reanalysis data from January, April, July, and October 2004 are
subsampled and interpolated onto the trajectory of AIRS nadir-view observations. Then
MODTRAN5 is used to generate synthetic AIRS radiances and synthetic spectral flux $F_{ERA}$. Then
$F_{AIRS-only}$ is derived from synthetic AIRS radiances based on the scene types determined from
synthetic AIRS radiances alone, instead of directly determined from ERA profiles as in our
previous works of Huang et al. (2008) and Chen et al. (2013). In total 290,761 profiles are used



and the number of profiles for each sub-scene type varies from 50 to 64992. The averaged
difference of the spectral flux for each scene type, denoted as $F_{AIRS-only}$ - $F_{ERA}$, at 10 cm$^{-1}$ spectral
interval is shown in Figure 2. For the window bands, the differences ($F_{AIRS-only}$ - $F_{ERA}$) are
generally within ± 0.03 Wm$^{-2}$ per 10 cm$^{-1}$. Exceptions are seen for those sub-scene types with
very dry atmosphere above a hot surface. These circumstances make it difficult for our
radiance-based algorithm to faithfully estimate the TPW. As shown in Table 3, the frequency of
occurrences for such scene types is small, e.g., hot surface with temperature above 310 K is no
more than 2%. Outside the window bands, the $F_{AIRS-only}$ - $F_{ERA}$ differences are usually within
±0.02 Wm$^{-2}$ per 10 cm$^{-1}$.
(2) For collocated AIRS and CERES clear-sky observations in 2004, we use the algorithm
to derive the spectral flux and OLR (the summation of spectral flux) from AIRS radiance
(hereafter, OLR$_{AIRS-only}$) and compare it with the collocated CERES clear-sky OLR (hereafter
OLR$_{CERES}$). Upper panels in Figure 3 show the annual-mean daytime and nighttime difference
between OLR$_{AIRS-only}$ and OLR$_{CERES}$, respectively. The differences are averaged onto 2° latitude by
2.5° longitude grids from 80°S to 80°N. Lower panels in Figure 3 show the histograms of OLR$_{AIRS-}$
$_{only}$-OLR$_{CERES}$ differences for all collocated AIRS and CERES clear-sky footprints. Figure 3a and 3b
show that the difference tends to be negative over land areas (~1-2 Wm$^{-2}$) and positive over
extra-tropical oceans (~1-3 Wm$^{-2}$). The RMS (root-mean-square) differences for Figure 3a and
3b are 1.79 and 1.11 Wm$^{-2}$, respectively. Such pattern and magnitude of the differences in
Figure 3a and 3b are comparable to the results using the scene type information directly from
the CERES SSF data set, as shown in Figure 5a and 5b in Chen et al. (2013). In terms of the
statistics of OLR$_{AIRS-only}$ − OLR$_{CERES}$ difference for individual footprint, the daytime mean





difference is 0.91 $Wm^{-2}$ with a standard deviation of 2.34 $Wm^{-2}$ (Figure 3c) and the nighttime
mean difference is 0.14 $Wm^{-2}$ with a standard deviation of 1.85 $Wm^{-2}$ (Figure 3d). These
statistics are comparable to those in Huang et al. (2008) and Chen et al. (2013).

(3) We examine the statistics of $OLR_{AIRS-only}$ - $OLR_{CERES}$ differences for each available clear-

sky sub-scene type in the data used in part (2). The results are summarized in Figure 4. The
averaged daytime $OLR_{AIRS-only}$ - $OLR_{CERES}$ differences for all sub-scene types are between -1.6
$Wm^{-2}$ and 3.3 $Wm^{-2}$ with a standard deviation no larger than 3.8 $Wm^{-2}$. For the nighttime, the
mean difference for all sub-scene types varies from -0.7 $Wm^{-2}$ to 2.2 $Wm^{-2}$ and the standard
deviation is less than 2.5 $Wm^{-2}$. Given that the radiometric uncertainty of CERES OLR is about 1%
and typical OLR value varies between 200-300 $Wm^{-2}$, the mean differences (black line in Figure
4) are within or at least comparable to the radiometric uncertainty of CERES OLR (red line in
Figure 4).

(4) In addition to using collocated clear-sky observations to evaluate the algorithm, we

also apply the algorithm to all collocated AIRS and CERES nadir-view observations in the entire
year of 2004 and obtain OLR for all AIRS measurements that our algorithm determines to be
clear-sky observations. The mean difference is 1.52 $Wm^{-2}$ and standard deviation is 2.46 $Wm^{-2}$.
The figure is not shown here. We then compare the OLR of those "false positive" observations,
i.e. footprints identified as clear-sky scenes by our algorithm but as cloudy-sky scenes by the
CERES algorithm. Figure 5 shows the histograms of OLR differences ($OLR_{AIRS-only}$ − $OLR_{CERES}$) of
such cases of "false positive". The mean difference is 2.93 $Wm^{-2}$ and 1.60 $Wm^{-2}$ for the daytime
and nighttime, respectively. The standard deviation is 2.3 $Wm^{-2}$ for both cases. The mean OLR
for the cases shown in Figure 5a and 5b is 288.7 $Wm^{-2}$ and 279.0 $Wm^{-2}$, respectively, which





means the relative difference between OLR$_{AIRS-only}$ and OLR$_{CERES}$ is only 1.0% and 0.6%. This
suggests that, even though the algorithm misclassifies such cloudy-sky observations as clear-sky
ones, the estimated OLR difference between OLR$_{AIRS-only}$ and OLR$_{CERES}$ is only 1% or less.
**5. Applicability to other viewing zenith angles (VZAs)**

The algorithm described above is for nadir-view AIRS radiances. It can be extended to

other viewing zenith angles by taking the dependency of upwelling radiances on viewing zenith
angles into account. Specifically, for the first two steps depicted in Section 3, the threshold
values and look-up-tables need to be adjusted in accordance with the viewing zenith angles.
The algorithm in the third component has already taken viewing zenith angle into account
(Huang et al., 2008) and thus no additional effort is needed. Since the objective of this study is
to demonstrate the feasibility of the algorithm, we summarize the performance of the
algorithm for other VZAs instead of describing all details as done for the case of nadir-view
observations. Figure 6a shows the success rate for the algorithm to accurately classify cloudy
and clear-sky footprints as a function of the VZA, which still uses the collocated CERES scene
type information as ground truth. The algorithm performs consistently across all VZAs; when
the VZA increases from zero to 45°, the success rate varies within 2%. Figure 6b shows the
differences between OLR$_{AIRS-only}$ and OLR$_{CERES}$ for both daytime and nighttime results. Both
differences, 1.93-2.15 Wm$^{-2}$ for daytime and 1.07-1.67 Wm$^{-2}$ for nighttime, change little with
respect to the VZA.

The performance with respect to different VZAs here is consistent with previous results

in Huang et al. (2008) and Chen et al. (2013), two studies that rely on the sub-scene type



information from the CERES SSF dataset. The algorithm in this study behaves robustly across
the range of VZAs for AIRS measurements. The other hyperspectral sounders make
observations over the similar range of VZAs. Therefore, the robust performances here further
assure the potential of extending the algorithm to other hyperspectral sounding observations.
**6. Conclusions and discussion**

Using AIRS observation as an example, this study develops an algorithm based solely on

spectral radiances to estimate LW clear-sky spectral flux. The algorithm first detects clear-sky
spectrum by a three-step threshold test, i.e., the "Golden Arches" test for the spatial
homogeneity, a bi-spectral test for spectral features of clear-sky absorption and emission, and a
single-channel thermal threshold test for an extra check against surface temperature.
Atmospheric and surface parameters (total precipitable water, lapse rate and surface
temperature) needed for categorizing sub-scene types are directly estimated using AIRS
radiances at six channels and the pre-constructed lookup tables. The accuracy of clear-sky
detection and sub-scene type classification, and their effect on clear-sky spectral flux derivation
have been assessed. When using CERES scene type information as the ground truth, the
algorithm can achieve an accuracy rate of 88.7% for classifying nadir-view clear-sky and cloudy
footprints. Differences between OLR derived using the algorithm and the collocated CERES OLR
show no strong dependence on the sub-scene types. The statistics of $OLR_{AIRS-only}$ - $OLR_{CERES}$
obtained here are comparable to those in Huang et al. (2008) and Chen et al. (2013), two
studies that directly used the scene-type and clear-sky information from the CERES data set.
The algorithm performs consistently over different viewing zenith angles.



The purpose of this study is to explore the additional value of hyperspectral sounding
measurements, i.e., by deriving spectral flux directly from such observations as the spectral
fluxes that have been shown to have unique merit in climate model evaluations (Huang et al.,
2006; Huang et al., 2013; Huang et al., 2014). The broadband flux measured by CERES and its
predecessor ERBE has become a benchmark standard in the earth observation community, so
does the sophisticated and well-validated multiple data-fusion approach used in the CERES data
product.  It is not the intention of this study to produce merely another set of broadband flux
products. Instead, the emphasis here is to derive the spectral flux, which can help us
understand the compensating biases in modeled broadband radiation flux.
In general, the performance of the algorithm is more affected by the accuracy of clear-
sky detection than the rest of components. To use LW spectral observations alone to detect
clear sky is not easy, partially because it is difficult to distinguish optically thin clouds or small
fraction of clouds within the field of view. In operational use, the accuracy of clear-sky
detection could be improved if other simultaneous measurements, especially those made at
higher spatial resolutions, are available. A good example is the use of MODIS imageries in the
CERES SSF algorithm. Another example is the use of microwave sounding observations to help
the surface parameter retrievals, which in turn helps the retrievals of atmospheric parameters
including the cloud vs. clear-sky detection (Kahn et al., 2014).
While the algorithm presented in this study is only for clear-sky spectra, it is conceivable
that this algorithm can be evolved for estimating spectral fluxes from cloudy-sky hyperspectral
observations as well. In the case of cloudy-sky spectra, the cloud parameters, especially cloud
fraction and cloud top height, will need to be considered in the definition of sub-scene types.





The rich information contained in hyperspectral radiances is likely sufficient to define sub-scene
types needed for the algorithm.
**Acknowledgments**

The AIRS Level 1B data are downloaded from NASA GSFC DAAC and the Aqua CERES

data were obtained from NASA Langley DAAC. The ECMWF ERA-interim data are from
http://data-portal.ecmef.int/data/d/. This research is supported by NASA under Grants
NNX14AJ50G and NNX15AC25G awarded to the University of Michigan.



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



Table 1. Threshold values used in the clear-sky tests. Details of threshold definitions and the
ways to determine them can be found in Section 3.1.

| Thresholds | Daytime Ocean | Nighttime Ocean | Daytime Land | Nighttime Land |
|---|---|---|---|---|
| C1 (K) | 0.62 | 0.61 | 2.17 | 1.650 |
| C2 (K) | -1.39 | -1.38 | -2.04 | -0.510 |
| C3 (K) | 2.47 ($Ts_{ERA}$<280 K) | 2.29($Ts_{ERA}$<280 K) | 1.24 ($Ts_{ERA}$<290 K) | 2.28 ($Ts_{ERA}$<260 K) |
|  | 3.12 (280-285 K) | 3.12 (280-285 K) | 1.49 (290-295 K) | 5.41 (260-270 K) |
|  | 3.61 (285-290 K) | 3.11 (285-290 K) | 3.28 (295-300 K) | 5.61 (270-275 K) |
|  | 3.61 (290-295 K) | 3.54 (290-295 K) | 3.99 (300-305 K) | 6.72 (275 -280 K) |
|  | 3.95 (295-300 K) | 4.13 (295-300 K) | 5.31 (305 -310 K) | 7.36(280-285 K) |
|  | 5.49 (> 300 K) | 5.82(>300 K) | 5.76 (>310 K) | 8.25 (>285 K) |


Table 2. The performance of clear-sky detection algorithm. FN (false negative) is the percentage
of CERES clear-sky observations misclassified as cloudy sky by the algorithm. FP (false positive)
is the percentage of CERES cloudy-sky observations misclassified as clear sky by the algorithm.
Accuracy is the overall success rate compared to the CERES algorithm in terms of distinguishing
clear- vs. cloudy-sky observations. Steps 1-3 are defined in detail in Section 3.1.

|  | Ocean | | | Land | | | Near-globe (81°S-81°N) | | |
|---|---|---|---|---|---|---|---|---|---|
|  | FN (%) | FP (%) | Accuracy (%) | FN (%) | FP (%) | Accuracy (%) | FN (%) | FP (%) | Accuracy (%) |
| Step 1 | 4.8 | 19.7 | 81.3 | 6.2 | 33.1 | 71.1 | 5.4 | 22.4 | 79.1 |
| Steps 1+2 | 9.7 | 14.1 | 86.2 | 10.0 | 19.2 | 82.2 | 9.8 | 15.2 | 85.3 |
| Steps 1+2+3 | 13.9 | 10.0 | 89.8 | 14.0 | 15.4 | 84.8 | 13.9 | 11.1 | 88.7 |




Table 3. Accuracy of the sub-scene type classification algorithm described in subsection 3.2. The
statistics are based on collocated nadir-view AIRS and CERES observations in 2004. 'Occ.' and
'Acc.' in the Table denotes occurrence and accuracy, respectively. The sub-scene type is coded
as a three-digit number. The first digit refers to TPW, the second one refers to $\Delta T$, and the last
refers to $T_s$, as defined in the table. The definition of sub-scene types here is identical to the LW
discrete intervals in Loeb et al. (2005).

| Sub-scene type | TPW (cm) | Occ. (%) | Acc. (%) | Sub-scene type | $\Delta T$ (K) | Occ. (%) | Acc. (%) | Sub-scene type | $T_s$ (K) | Occ. (%) | Acc. (%) |
|---|---|---|---|---|---|---|---|---|---|---|---|
| 1-- | 0-1 | 16.3 | 63.1 | -1- | <15 | 32.9 | 70.5 | --1 | <270 | 1.24 | 99.8 |
| 2-- | 1-3 | 55.0 | 86.8 | -2- | 15-30 | 65.8 | 85.1 | --2 | 270-290 | 24.7 | 98.2 |
| 3-- | 3-5 | 25.7 | 82.0 | -3- | 30-45 | 1.29 | 48.4 | --3 | 290-310 | 73.1 | 93.2 |
| 4-- | >5 | 3.0 | 53.8 | -4- | >45 | 0.002 | 16.7 | --4 | 310-330 | 0.98 | 22.1 |
| | | | | | | | | --5 | >330 | 0.0 | - |
| Overall | | 100 | 80.7 | | | 100 | 79.8 | | | 100 | 93.8 |





488                                    **Figure Captions**

Figure 1. Histogram of the standard deviations of 963.8 cm$^{-1}$ brightness temperatures among an
AIRS clear-sky footprint and four adjacent AIRS footprints derived. The clear-sky information
from collocated CERES observation is used. The histograms for daytime ocean, daytime land,
nighttime ocean, and nighttime land are plotted separately. The black dash line denotes the 95%
percentile and corresponds to the value of C1 shown in Table 1.
Figure 2. The mean differences between the predicted spectral fluxes based on synthetic AIRS
spectra and the directly computed fluxes for different sub-scene types. The naming convention
of sub-scene type is defined in Table 3. The spectral flux is for every 10 cm$^{-1}$ interval from 10 cm$^{-}$
$^{1}$ to 2000 cm$^{-1}$.
Figure 3. (a) Near-global distribution of annual-mean differences between daytime OLR derived
from clear-sky AIRS nadir-view radiances using the algorithm in this study and the collocated
CERES clear-sky daytime OLR (OLR$_{AIRS-only}$ - OLR$_{CERES}$). The data in 2004 is used and averaged onto
2.5° longitude by 2° latitude grids. (b) Same as (a) but for annual-mean nighttime OLR
differences. (c) The histograms of daytime OLR$_{AIRS-only}$ - OLR$_{CERES}$ differences among all collocated
AIRS and CERES nadir-view footprints. (d) Same as (c) but for the histogram of nighttime
OLR$_{AIRS-only}$ - OLR$_{CERES}$ differences. Fifty bins are used in both (c) and (d). The mean differences ±
standard deviations and number of observations are also labeled on the plot.
Figure 4. (a) Black line denotes the mean of daytime OLR difference (OLR$_{AIRS-only}$ - OLR$_{CERES}$) for
each sub-scene type. Ticked vertical lines denote ±1σ (standard deviation). Red line is the
uncertainty of OLR$_{CERES}$ (assuming 1% of mean OLR$_{CERES}$ for all scene types). Blue bars indicate
the frequency of occurrence of each sub-scene type in percentage. (b) Same as (a) but for





nighttime observations. The numbers of daytime and nighttime observations are $1.86 \times 10^5$ and
$1.87 \times 10^5$, respectively.
Figure 5. (a) and (b) are similar as Figure 3(c) and 3 (d) but for the AIRS footprints classified as
clear sky by the algorithm in this study while their collocated CERES footprints are identified as
cloudy sky. Mean ± standard deviation of the difference ($OLR_{AIRS-only} - OLR_{CERES}$) is also given on
the plot.
Figure 6. (a) Success rate of the algorithm in distinguishing clear-sky and cloudy-sky footprints
as a function of viewing zenith angle (VZA). (b) The difference of $OLR_{AIRS-only} - OLR_{CERES}$ as a
function of VZA. Ticked vertical lines denote the $\pm 1\sigma$ (standard deviation).



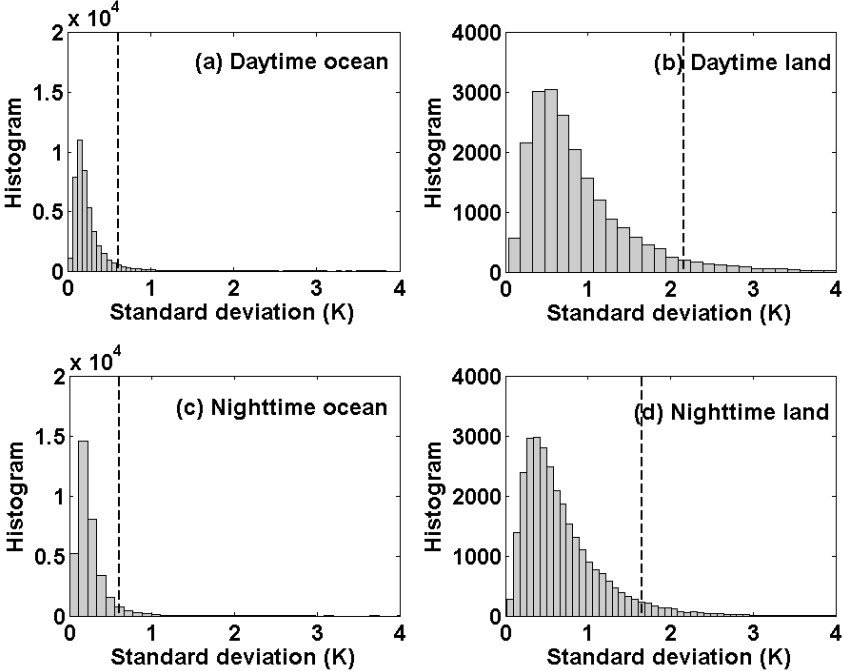


Figure 1. Histogram of the standard deviations of 963.8 cm[-1] brightness temperatures among an

AIRS clear-sky footprint and four adjacent AIRS footprints derived. The clear-sky information

from collocated CERES observation is used. The histograms for daytime ocean, daytime land,

nighttime ocean, and nighttime land are plotted separately. The black dash line denotes the 95%

percentile and corresponds to the value of C1 shown in Table 1.





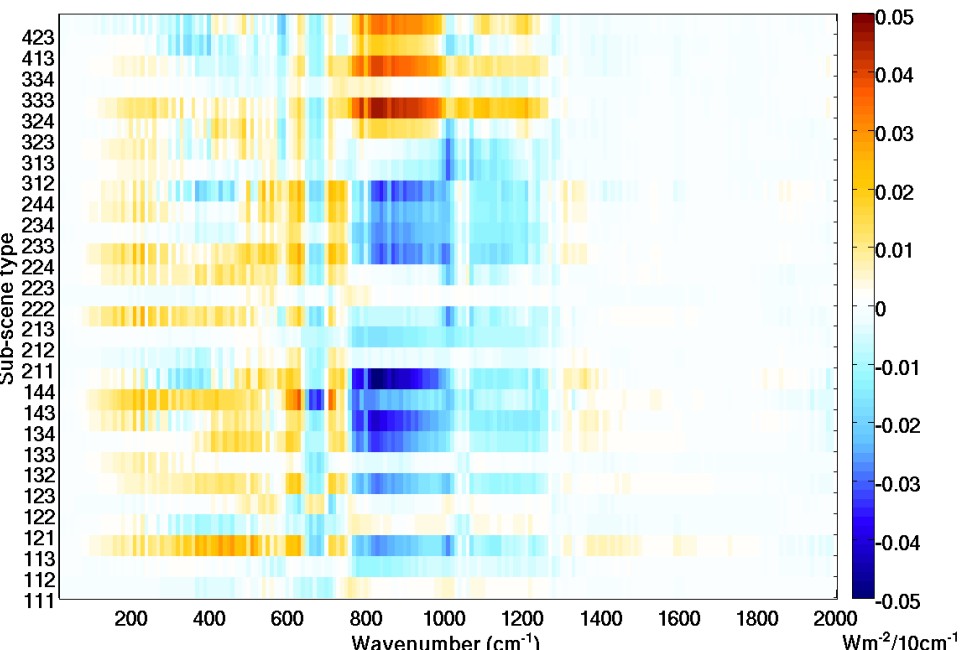

526

Figure 2. The mean differences between the predicted spectral fluxes based on synthetic AIRS

spectra and the directly computed fluxes for different sub-scene types. The naming convention

of sub-scene type is defined in Table 3. The spectral flux is for every 10 cm$^{-1}$ interval from 10 cm$^{-}$

$^1$ to 2000 cm$^{-1}$.





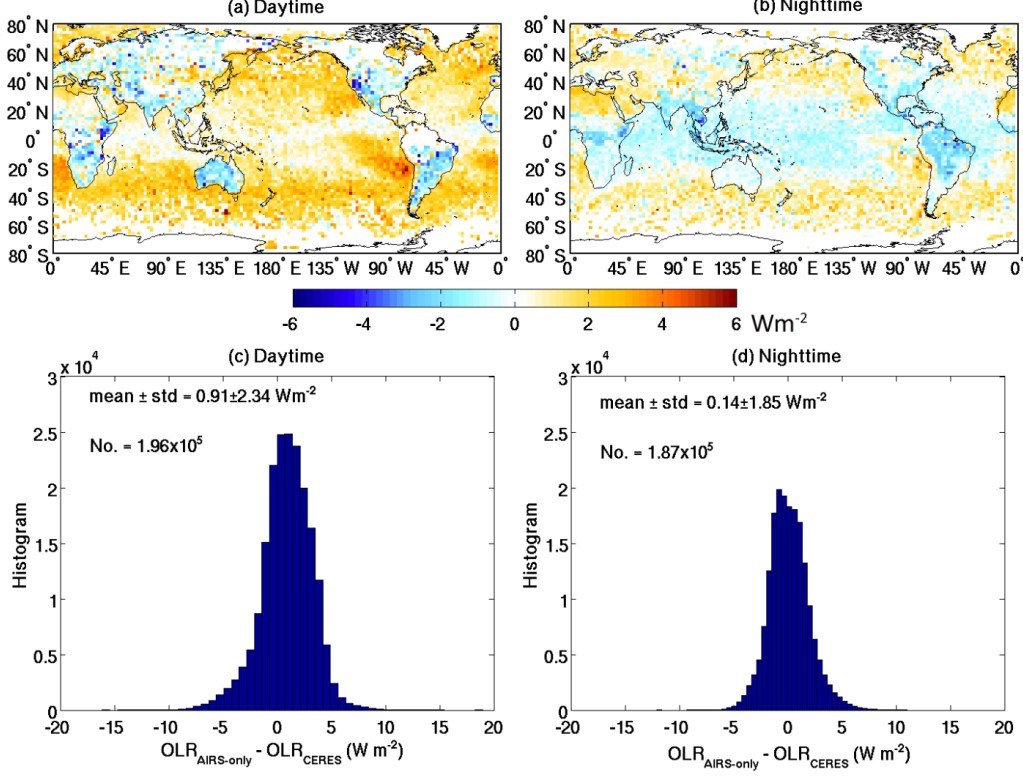


Figure 3. (a) Near-global distribution of annual-mean differences between daytime OLR derived
from clear-sky AIRS nadir-view radiances using the algorithm in this study and the collocated
CERES clear-sky daytime OLR ($OLR_{AIRS-only}$ - $OLR_{CERES}$). The data in 2004 is used and averaged onto
2.5° longitude by 2° latitude grids. (b) Same as (a) but for annual-mean nighttime OLR
differences. (c) The histograms of daytime $OLR_{AIRS-only}$ - $OLR_{CERES}$ differences among all collocated
AIRS and CERES nadir-view footprints. (d) Same as (c) but for the histogram of nighttime
$OLR_{AIRS-only}$ - $OLR_{CERES}$ differences. Fifty bins are used in both (c) and (d). The mean differences ±
standard deviations and number of observations are also labeled on the plot.



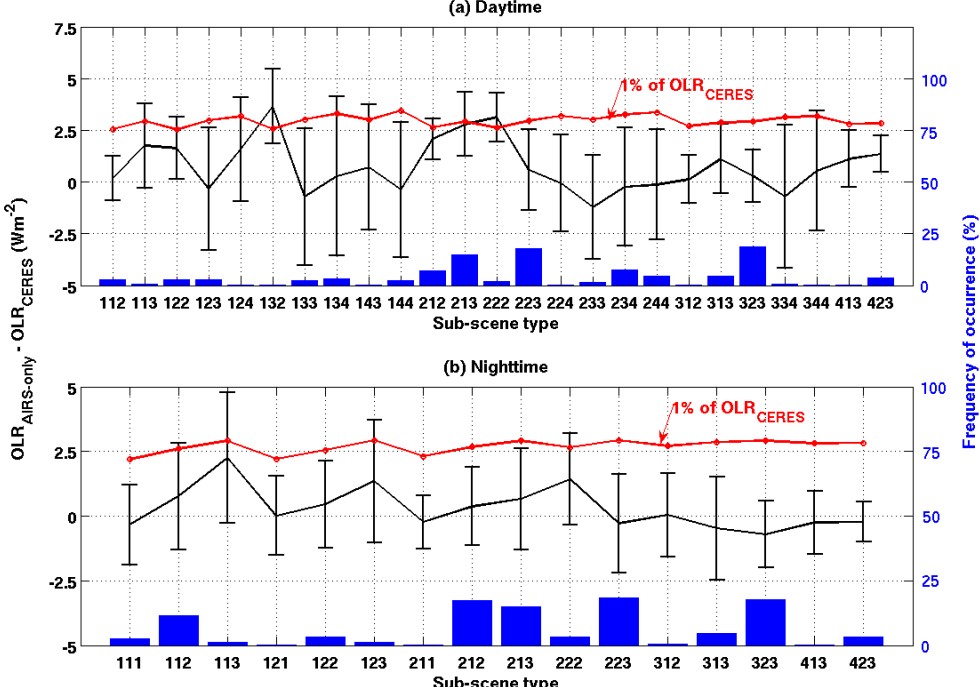


Figure 4. (a) Black line denotes the mean of daytime OLR difference ($OLR_{AIRS\text{-}only}$ - $OLR_{CERES}$) for
each sub-scene type. Ticked vertical lines denote ±1σ (standard deviation). Red line is the
uncertainty of $OLR_{CERES}$ (assuming 1% of mean $OLR_{CERES}$ for all scene types). Blue bars indicate
the frequency of occurrence of each sub-scene type in percentage. (b) Same as (a) but for
nighttime observations.  The numbers of daytime and nighttime observations are $1.86\times10^5$ and
$1.87\times10^5$, respectively.



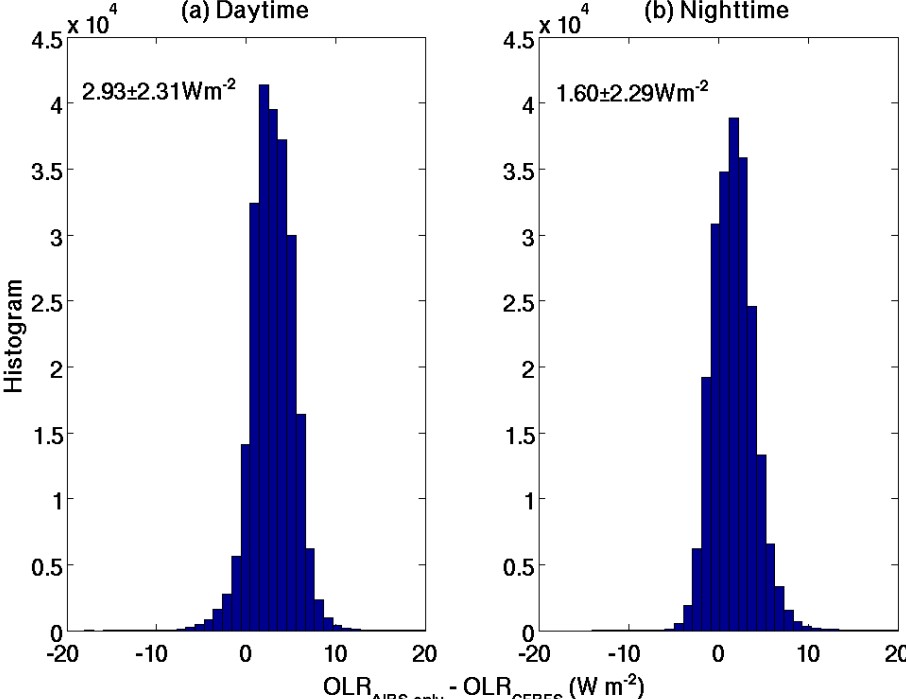


Figure 5. (a) and (b) are similar as Figure 3(c) and 3 (d) but for the AIRS footprints classified as

clear sky by the algorithm in this study while their collocated CERES footprints are identified as

cloudy sky.  Mean ± standard deviation of the difference ($OLR_{AIRS-only} - OLR_{CERES}$) is also given on

the plot.







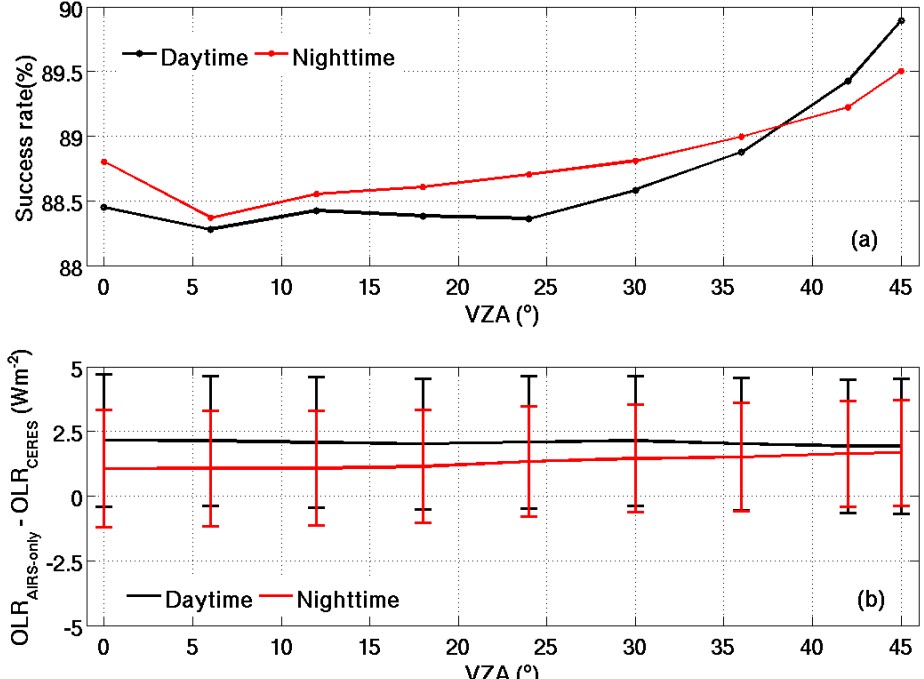


Figure 6. (a) Success rate of the algorithm in distinguishing clear-sky and cloudy-sky footprints
as a function of viewing zenith angle (VZA). (b) The difference of $OLR_{AIRS-only} - OLR_{CERES}$ as a
function of VZA. Ticked vertical lines denote the $\pm 1\sigma$ (standard deviation).