# Peer review of "Deriving clear-sky longwave spectral flux solely from hyperspectral radiance: a case study with AIRS observations"

_Atmospheric Measurement Techniques, 2016_

## Referee Comment (RC1) · Anonymous Referee #1 · 9 Oct 2016

The manuscript titled "Deriving clear-sky longwave spectral flux solely from hyperspectral radiance: a case study with AIRS observations" by X. Chen, and X. Huang contains significant improvement over their previously developed algorithms for deriving spectral flux from hyperspectral radiances. The work is important since the spectral flux can provide important information for the climate model diagnostics. The main innovation of this study is that the TOA flux can be derived without using external satellite data. In their previous method, the radiance-to-flux conversion uses scene type information derived from other collocated satellite observations. Since the hyperspectral radiance spectra contain information on atmospheric temperature, water vapor, and clouds, it is possible to identify the scene types and estimate the spectral flux directly from the
hyperspectral radiances without using auxiliary information from other satellites. The accuracy of the proposed method has been demonstrated using both synthetic data and AIRS observations. The results show that the mean differences in the spectral fluxes obtained from radiance-to-flux method and those computed directly from ERA profiles using the MODTRAN radiative transfer model are small (about 0.03 W/m2). The RMS differences between the AIRS-only calculated OLR and the CERES clear-sky OLR are comparable to their previous method. The averaged OLR differences are comparable or less than the radiometric uncertainty of CERES OLR. Even for the misclassified sub-scene type, the error in the predicted OLR is only 1% or less. The robust performance of this technique at different viewing zenith angles indicates the success of this new approach. Thus, I suggest that Atmospheric Measurement Techniques to publish this work. The manuscript is well written; the reviewer only has a few suggestions for the authors:

1. The authors used a spatial inhomogeneity test, a bi-spectral test, and a thermal threshold test to verify a clear-sky spectrum. The accuracy is around 80-90%. What is the rationale for using 4 adjacent footprints for the spatial inhomogeneity test? Since the AIRS field of regard has 9 footprints, it makes more sense to use all 8 footprints around the center one. It will probably increase the sensitivity of the inhomogeneity test while not increasing the area size used for the test. 2. On page 12 line 211, change the "ADM (Rairs)" to "ADM, Rairs,"

---

## Referee Comment (RC2) · Anonymous Referee #2 · 9 Oct 2016

**GENERAL COMMENTS**

The manuscript addresses a relevant technical question (radiance-to-flux conversion) in the retrieval process of products from AIRS satellite instrument data and instruments alike. As to my knowledge, the presented algorithm is a unique realisation of such a radiance-to-flux conversion, even though the concept of "deriving flux solely from radiance" is not new (e.g. doi:10.1007/s00376-015-5040-8). Since the derivation of the flux from such measurements generally contains several assumptions and is, by no means, straight-forward, I support a diversity of reported conversion algorithms.

The conclusion, i.e. the presented algorithm compares well with CERES OLR estimates, is sufficient for publication. The methods and assumptions are mostly sound

and clear; see "specific comments" for limitations. The manuscript is well structured and language is fluent and precise.

In the following, I will only address aspects listed at http://www.atmospheric-measurement-techniques.net/peer_review/review_criteria.html, if they give reason for concerns.

SPECIFIC COMMENTS

One major issue that concerns me is the clear-sky selection. It should be clearly outlined, why the algorithm is refined to clear-sky cases at all. Hints are given the very last paragraph of the manuscript. I suggest to add appropriate reasoning in the introduction around line 74; a short mentioning of the reason in the abstract could also be helpful.

The clear-sky selection process as described in section 3.1.1 of the manuscript seams reasonable. However, the assessment of the performance has substantial deficits:

The authors use "accuracy" as metric for the performance, which they define as "percentage of cases in which our algorithm can correctly classify the footprints". Given that only 9-10% of all observations are clear-sky (line 185, I assume this is the figure for CERES classification), there is no skill needed to reach an overall accuracy of 88.7% (Table 2: Near-globe Accuracy of Steps 1+2+3). Using the trivial approach "all observations are cloudy-sky" would yield an accuracy of 90-91%!

Furthermore, the "false positive" metric and the implications of the according numbers given in Table 2 is not addressed well enough in the performance assessment (section 3.1.2). The given false positive rate of 11.1% (Table 2: Near-globe FP of Steps 1+2+3) actually exceeds the occurence of the event (i.e. 9-10% clear-sky).

Say there were 1000 CERES cloudy-sky observations and 100 clear-sky:

FP=11.1% yields 111 AIRS clear-sky are CERES cloudy-sky (misclassified) and 889 AIRS cloudy-sky are CERES cloudy-sky (correct)

[Figure]

FN=13.9% yields ∼14 AIRS cloudy-sky are CERES clear-sky (misclassified) and ∼86 AIRS clear-sky are CERES clear-sky (correct)

In total, this is:

889+86 = 975 correct classifications (consistent with ACC=88.7% of 1100 observations)

111+86 = 197 AIRS clear-sky classifications

889+14 = 903 AIRS cloudy-sky classifications

So, essentially more than half of the AIRS spectra for which the radiance-to-flux algorithm shall be applied, are "misclassified" in the CERES-comparison-sense. I would not say that this rules out the suitability of the clear-sky detection algorithm. Also, most CERES clear-sky observations are captured by the detection algorithm, which proves skill. Since the CERES clear-sky fraction does not compare well with an average planetary cloud-cover of around 2/3, while the suggested AIRS-based detection comes closer to that, this needs some discussion in the assessment. Footprint-size is definitely an issue here, since classification seems to sort out pixels with very little cloud cover amount inside the pixel.

The reader should be clearly informed that the number of clear-sky observations used for AIRS- and CERES-derived flux comparison differs by a factor greater than two. Since this fact does not become apparent from the presented data in section 3.1.2 and Table 2, I would argue, that the applied metrics (FN, FP, Accuracy) are rather inappropriate and misleading here. Presenting and discussing actual numbers or other metrics and a more comprehensive discussion of this issue would improve the manuscript. Another possibility to improve comparability of results with CERES could be stricter threshold values for clear-sky detection.

I see a similar problem in the assessment of sub-scene type classification (section 3.2). The authors summarise, that the accuracy of their classification is "80% or even

higher" (line 204). I cannot support this conclusion from table 3, since common sub-scene types (e.g. "-1-") have substantially lower accuracy. More important though, the "accuracy" metric does not seem appropriate to judge the performance. Many sub-scene types in table 3 have a lower occurence than the misclassification rate (by which I mean the complement to "accuracy").

Minor comments:

* Mentioning of the word "satellite" in title, abstract, and/or key words would help readers to assign the article to the right discipline

* The term "solely" in title and abstract is slightly misleading, since ECMWF surface temperature analysis is used in the process (line 150)

* Mention footprint size/ pixel size of CERES SSF detection of clear-sky in section 2

* Does the algorithm account for attenuation due to the height of the satellite above the "top of atmosphere"?

* line 211: Is F_AIRS an integration over Theta? This is missing in the equation.

* line 212-213: Please provide a reference to the exact algorithm used. E.g., which spectral range is used? From lines 97-98 one could assume a greater spectral range than what Huang et al. (2008) uses.

* line 236: Reasoning why "These circumstances make it difficult" would be helpful here

TECHNICAL CORRECTIONS/ COMMENTS

* The paper uses wavelength and wavenumber for referring to certain parts of the IR-spectrum. This seems particularly confusing in two paragraphs: lines 96-101 and 135-156. Using one of the two consistently would improve readability.

* line 105, line 391-394: I could not find the referenced document, only a "Version

1" of it under http://ceres.larc.nasa.gov/collect_guide.php. Please provide an URL if possible.

* line 159: consider changing "another words" to "other words"

* line 168: consider changing "dash-dot" to "dashed"

* Figure 1: Equal bin sizes in all panels would improve comparability

* line 239: From Table 3, I would read "no more than 1%", not 2% (sum of sub-scene types "–4" and "–5")

* line 240 and Figure 2: These numbers are hard to read from the figure. Consider using a discrete colourbar, e.g. in steps of 0.005 Wm-2/10cm-1

* Figure 2: "Sub-scene type" labels should be moved slightly up to align with the corresponding colouring in the figure

* Figure 2: Marking the spectral bands of actual AIRS observations in the figure would help in judging the differences shown. This could easily be marked on the top or bottom axis.

* line 250: Are the numbers for RMS weighted by area of the respective $2°x2.5°$-pixel?

* line 270-273: The reader gets the impression, that only the "false positive" cases were used here, but lines 268-269 state otherwise (all AIRS clear-sky, i.e. "false and correct positive"). Please clarify.

---

## Author Comment (AC1) · 18 Nov 2016

We are thankful to the reviewer for his or her thoughtful comments, which improve the clarity of our manuscript. All the points by the reviewer are well taken, and we have made revisions to the paper to clarify or address the points raised by the referee. We will first reproduce the referee's questions with italic font in quote, followed by our responses in blue font (so responses can be easily separated from the referee's questions). Both authors listed on the manuscript have concurred with submission in this revised form.

"*The manuscript titled "Deriving clear-sky longwave spectral flux solely from hyperspectral radiance: a case study with AIRS observations" by X. Chen, and X. Huang contains significant improvement over their previously developed algorithms for deriving spectral flux from hyperspectral radiances. The work is important since the spectral flux can provide important information for the climate model diagnostics. The main innovation of this study is that the TOA flux can be derived without using external satellite data. In their previous method, the radiance-to-flux conversion uses scene type information derived from other collocated satellite observations. Since the hyperspectral radiance spectra contain information on atmospheric temperature, water vapor, and clouds, it is possible to identify the scene types and estimate the spectral flux directly from the hyperspectral radiances without using auxiliary information from other satellites. The accuracy of the proposed method has been demonstrated using both synthetic data and AIRS observations. The results show that the mean differences in the spectral fluxes obtained from radiance-to-flux method and those computed directly from ERA profiles using the MODTRAN radiative transfer model are small (about 0.03 W/m2). The RMS differences between the AIRS-only calculated OLR and the CERES clear-sky OLR are comparable to their previous method. The averaged OLR differences are comparable or less than the radiometric uncertainty of CERES OLR. Even for the misclassified sub-scene type, the error in the predicted OLR is only 1% or less. The robust performance of this technique at different viewing zenith angles indicates the success of this new approach. Thus, I suggest that Atmospheric Measurement Techniques to publish this work. The manuscript is well written; the reviewer only has a few suggestions for the authors:.*"

We thank the reviewer for the summary and positive assessment. Since this is a general overview comment, we have no response.

1. "*The authors used a spatial inhomogeneity test, a bi-spectral test, and a thermal threshold test to verify a clear-sky spectrum. The accuracy is around 80-90%. What is the rationale for using 4 adjacent footprints for the spatial inhomogeneity test? Since the AIRS field of regard has 9 footprints, it makes more sense to use all 8 footprints around the center one. It will probably*"

*increase the sensitivity of the inhomogeneity test while not increasing the area size used for the test.*"

As illustrated in the figure below, we assume footprint #5 is the footprint to be detected, and we used 5 footprints (#2, #4, #5, #6 and #8) to calculate standard deviation in our study. Using 5 footprints can guarantee footprint 5 to be clear-sky if footprints 2, 4, 6 and 8 are clear-sky (i.e., the standard deviation is very small). If we use all 9 footprints, footprint #5 might be incorrectly flagged as cloudy-sky when one or more footprints from #1, #3, #7 and #9 is cloudy (i.e., the standard deviation increase) even the rest footprints (#2, #4, #6 and #8) are clear-sky.

[Figure]

Figure 1. Illustration of the footprint #5 , the footprint to be detected and its eight adjacent footprints in an AIRS field of regard. Footprint #7 here is covered by cloud.

2. "*On page 12 line 211, change the "ADM (Rairs)" to "ADM, Rairs,"*."

We changed it as the reviewer suggested.

Again, we are sincerely thankful to the reviewer for the thoughtful comments which have greatly improved the quality and presentation of our manuscript. We have acknowledged the reviewer in the text for the efforts.

Xiuhong Chen and Xianglei Huang

---

## Author Comment (AC2) · 18 Nov 2016

**Manuscript No. amt-2016-268 Reply Letter to Reviewer 2**

We are thankful to the reviewer for his or her thorough and thoughtful comments, which greatly improve the clarity of our manuscript. All the points by the reviewer are well taken, and we have made revisions to the paper to clarify or address the points raised by the referee. We will first reproduce the referee's questions with italic font in quote, followed by our responses in blue font (so responses can be easily separated from the referee's questions). Both authors listed on the manuscript have concurred with submission in this revised form.

**GENERAL COMMENTS**

"*The manuscript addresses a relevant technical question (radiance-to-flux conversion) in the retrieval process of products from AIRS satellite instrument data and instruments alike. As to my knowledge, the presented algorithm is a unique realisation of such a radiance-to-flux conversion, even though the concept of "deriving flux solely from radiance" is not new (e.g. doi:10.1007/s00376-015-5040-8). Since the derivation of the flux from such measurements generally contains several assumptions and is, by no means, straight-forward, I support a diversity of reported conversion algorithms.*

*The conclusion, i.e. the presented algorithm compares well with CERES OLR estimates, is sufficient for publication. The methods and assumptions are mostly sound and clear; see "specific comments" for limitations. The manuscript is well structured and language is fluent and precise. In the following, I will only address aspects listed at http://www.atmosphericmeasurement-techniques.net/peer_review/review_criteria.html, if they give reason for concerns.*"

We thank the reviewer for positive assessment of our study. Since this is a general overview comment, we have no response to it.

**SPECIFIC COMMENTS**

1. "*One major issue that concerns me is the clear-sky selection. It should be clearly outlined, why the algorithm is refined to clear-sky cases at all. Hints are given the very last paragraph of the manuscript. I suggest to add appropriate reasoning in the introduction around line 74; a short mentioning of the reason in the abstract could also be helpful.*"

The primary motivation for us to focus on clear-sky cases in this study is that classifying clear-sky sub-scene types is much simpler than classifying cloudy sub-scene types, given the difficulty involved in estimating cloud macroscopic properties. Thus we started this series of work from the simple case first.

Following the reviewer's suggestion, we rewrote sentence in Line 75 (in the introduction section). Now it reads "As a first step, this study focuses on clear-sky scene types, i.e., only non-cloud parameters (precipitable water, lapse rate and surface temperature) are considered in the definition of sub-scene types".

We also updated the abstract to explain this as suggested by the reviewer. Now Line 6 reads " The identified clear-sky scenes are then categorized into different sub-scene types using information about precipitable water, lapse rate and surface temperature inferred from the AIRS radiances at six selected channels.".

2. *"The clear-sky selection process as described in section 3.1.1 of the manuscript seams reasonable. However, the assessment of the performance has substantial deficits:*

*The authors use "accuracy" as metric for the performance, which they define as "percentage of cases in which our algorithm can correctly classify the footprints". Given that only 9-10% of all observations are clear-sky (line 185, I assume this is the figure for CERES classification), there is no skill needed to reach an overall accuracy of 88.7% (Table 2: Near-globe Accuracy of Steps 1+2+3). Using the trivial approach "all observations are cloudy-sky" would yield an accuracy of 90-91%!*

*Furthermore, the "false positive" metric and the implications of the according numbers given in Table 2 is not addressed well enough in the performance assessment (section 3.1.2). The given false positive rate of 11.1% (Table 2: Near-globe FP of Steps 1+2+3) actually exceeds the occurence of the event (i.e. 9-10% clear-sky).*"

We understand the reviewer's argument about the deficiency in the definition of "accuracy" in our metrics. In the revision, we followed Amato et al. (2014) to use Merit Function defined in it for the successful classifications of both clear-sky and cloudy-sky AIRS footprints. This information is now added into Section 3.1.2. It reads (Lines 189-190 in revised manuscript) "The definition of FN, FP and merit function follows Amato et al. (2014)." Actually, the values of merit function are the same as the values we obtained using the definition of "accuracy". We think the metrics of merit function (or the accuracy in our original manuscript), FN, FP altogether form the full perspective of the performance of the algorithm. It is true that no skills are needed for the merit function/accuracy to yield 91% accuracy as argued by the reviewer. However, it is also possible that an ill-designed algorithm can make the merit function even much lower than 91%.

As stated in Lines 345-347 in revision "To use LW spectral observations alone to detect clear sky is not easy, partially because it is difficult to distinguish optically thin clouds or small fraction of clouds within the field of view." To corroborate this statement, we add a new figure (Figure 2 in revision) to show that the dominant majority of cloudy-sky footprints misclassified as clear-sky scenes by our algorithm is those footprints with low cloud (cloud top pressure at ~900 hPa or even lower) and cloud fraction less than 10%. We also stated in Lines 193-196 (new

Line number in revision) "As far as the FN and FP rates are concerned, this algorithm is comparable to other clear-sky detection algorithms that are based on IR spectral radiances alone (e.g. Table 4 in Amato et al., 2014)".

In summary, we now used the same metrics as in Amato et al. (2014) and we added a new figure to show the cloudy-sky scenes that are misclassified as clear-sky scenes by our algorithm are mostly scenes with low clouds and a cloud fraction less than 10%. As shown in Figure 6, because of the low cloud top and small cloud fraction, the OLR estimated for these misclassified scenes indeed is not deviating from the OLR from collocated CERES results that much (the mean relative difference is no more than 1%).

Reference: Amato, U., Lavanant, L., Liuzzi, G., Masiello, G., Serio, C., Stuhlmann, R. and Tjemkes, S. A.: Cloud mask via cumulative discriminant analysis applied to satellite infrared observations: scientific basis and initial evaluation. Atmos Meas Tech, 7, 3355–3372, 2014.

3. *"Say there were 1000 CERES cloudy-sky observations and 100 clear-sky:*

*FP=11.1% yields 111 AIRS clear-sky are CERES cloudy-sky (misclassified) and 889*

*AIRS cloudy-sky are CERES cloudy-sky (correct)*

*FN=13.9% yields ~14 AIRS cloudy-sky are CERES clear-sky (misclassified) and ~86*

*AIRS clear-sky are CERES clear-sky (correct)*

*In total, this is:*

*889+86 = 975 correct classifications (consistent with ACC=88.7% of 1100 observations)*

*111+86 = 197 AIRS clear-sky classifications*

*889+14 = 903 AIRS cloudy-sky classifications*

*So, essentially more than half of the AIRS spectra for which the radiance-to-flux algorithm shall be applied, are "misclassified" in the CERES-comparison-sense. I would not say that this rules out the suitability of the clear-sky detection algorithm. Also, most CERES clear-sky observations are captured by the detection algorithm, which proves skill. Since the CERES clear-sky fraction does not compare well with an average planetary cloud-cover of around 2/3, while the suggested AIRS-based detection comes closer to that, this needs some discussion in the assessment. Footprint-size is definitely an issue here, since classification seems to sort out pixels with very little cloud cover amount inside the pixel.*

*The reader should be clearly informed that the number of clear-sky observations used for AIRS- and CERES-derived flux comparison differs by a factor greater than two. Since this fact does not become apparent from the presented data in section 3.1.2 and Table 2, I would argue, that the*

*applied metrics (FN, FP, Accuracy) are rather inappropriate and misleading here. Presenting and discussing actual numbers or other metrics and a more comprehensive discussion of this issue would improve the manuscript. Another possibility to improve comparability of results with CERES could be stricter threshold values for clear-sky detection.*

*I see a similar problem in the assessment of sub-scene type classification (section 3.2). The authors summarise, that the accuracy of their classification is "80% or even higher" (line 204). I cannot support this conclusion from table 3, since common subscene types (e.g. "-1-") have substantially lower accuracy. More important though, the "accuracy" metric does not seem appropriate to judge the performance. Many subscene types in table 3 have a lower occurence than the misclassification rate (by which I mean the complement to "accuracy").”*

As the reviewer pointed out, the clear-sky fraction of a space-borne instrument is dependent on the field of view of the instrument as well as the definition of clear-sky scene by its algorithm. To make the readers better informed, we added one sentence in Section 2 (Lines 110-111 in the revised manuscript) to explain this. It reads "A CERES field of view is classified as a clear-sky scene if the coincident MODIS pixel-level cloud coverage within the FOV is less than 0.1%”.

We adopted the reviewer's suggestion and add sentences to the end of Section 3.1.2 to discuss about the misclassification here in more details. It reads: "The number of clear-sky AIRS footprints detected by this algorithm is nearly twice as many as the number of clear-sky footprints from collocated CERES SSF data set. The overwhelming majority of misclassified footprints are those with cloud top pressure $>=$ 900 hPa and cloud fraction $<=10\%$, as shown in Figure 2. In another word, for footprints with low cloud and very small cloud fraction, the IR-alone detection algorithm has difficulty to distinguish it from clear-sky footprint, which is consistent with previous IR-based clear-sky detection results. As shown later in Figure 6 and related discussion, the OLR estimated for these misclassified footprints are indeed similar to the collocated CERES OLR, largely because of the very limited cloud fraction in the footprints.”

**Minor comments:**

1. *"Mentioning of the word "satellite" in title, abstract, and/or key words would help readers to assign the article to the right discipline”*

We added the word of "space-borne" into the title, abstract and key words as the reviewer suggested.

2. *"The term "solely" in title and abstract is slightly misleading, since ECMWF surface temperature analysis is used in the process (line 150)”*

We removed the word of "solely" to avoid the confusion.

3. *"Mention footprint size/ pixel size of CERES SSF detection of clear-sky in section 2”*

We added the information as the reviewer suggested. It reads "CERES nadir-view field of view (FOV) is ~20 km at surface."

4. *" Does the algorithm account for attenuation due to the height of the satellite above the "top of atmosphere"?"*

The attenuation for the thermal infrared radiation above the "top of atmosphere" (which is usually set at 100 km in forward modeling) is little. Therefore, even the altitude for the satellite is ~800 km, it is a very common practice to have forward model only up to 100 km. We follow the same modeling approach. There is no need to consider radiative transfer above 100 km for thermal infrared, simply because of the tiny air mass and thus little absorption and emission above it.

5. *"line 211: Is F_AIRS an integration over Theta? This is missing in the equation."*

Yes. It is the spectral flux as usually defined. The equation is to use radiance and anisotropic factor, both of which are theta-dependent, to infer the spectral flux. No actual angular integration is performed. This is the standard approach for inferring flux from radiance observations as done by the ERBE and CERES algorithms, as well as in Huang et al. (2008) and follow-up studies. We rewrote this sentence with reference added to clarify this.

6. *"line 212-213: Please provide a reference to the exact algorithm used. E.g., which spectral range is used? From lines 97-98 one could assume a greater spectral range than what Huang et al. (2008) uses."*

We now rewrite the sentence. It reads "Then a principle component-based multivariate prediction scheme is used to estimate spectral fluxes over the spectral portion ($< 649.6$ cm$^{-1}$, $1136.6 - 1217.0$ cm$^{-1}$ and $1613.9 - 2000$ cm$^{-1}$) not covered by the AIRS instrument. Details can be found in Huang et al. (2008)."

This study used the same spectral range as that in Huang et al. (2008). Now we add one sentence behind Lines 97-98 in the original version. It reads "The near-IR band is not used to derive spectral fluxes over $10 - 2000$ cm$^{-1}$."

7. *"line 236: Reasoning why "These circumstances make it difficult" would be helpful here"*

In our algorithm, TPW is estimated based on a look-up-table using the method in Chen and Huang (2014). As the actual occurrence of dry atmosphere above a hot surface is not as frequent as that of other combinations, the look-up-table for this particular scene type was not trained as robustly as for other scene types. This consequently makes the estimate of TPW over such dry and hot scenes difficult.

**TECHNICAL CORRECTIONS/ COMMENTS**

1. *"The paper uses wavelength and wavenumber for referring to certain parts of the IR-spectrum. This seems particularly confusing in two paragraphs: lines 96-101 and 135-156. Using one of the two consistently would improve readability."*

We used wavenumber throughout the entire manuscript in the revision.

2. *" line 105, line 391-394: I could not find the referenced document, only a "Version 1" of it under http://ceres.larc.nasa.gov/collect_guide.php. Please provide an URL if possible.*

We thank the reviewer for catching this typo. It is Version 1 not Version 6. We corrected it and added the URL onto the reference.

3. *"line 159: consider changing "another words" to "other words""*

We changed it as the reviewer suggested.

4. *"line 168: consider changing "dash-dot" to "dashed""*

We changed as it the reviewer suggested.

5. *"Figure 1: Equal bin sizes in all panels would improve comparability"*

We changed it as the reviewer suggested.

6. *"line 239: From Table 3, I would read "no more than 1%", not 2% (sum of sub-scene types "–4" and "–5")"*

We thank the reviewer for catching this typo. We corrected it.

7. *"line 240 and Figure 2: These numbers are hard to read from the figure. Consider using a discrete colourbar, e.g. in steps of 0.005 Wm$^{-2}$/10cm$^{-1}$"*

We updated the plot as the reviewer suggested.

8. *"Figure 2: "Sub-scene type" labels should be moved slightly up to align with the corresponding colouring in the figure"*

We changed it as the reviewer suggested.

9. *" Figure 2: Marking the spectral bands of actual AIRS observations in the figure would help in judging the differences shown. This could easily be marked on the top or bottom axis."*

We changed it as the reviewer suggested.

10. *"line 250: Are the numbers for RMS weighted by area of the respective 2x2.5-pixel?"*

Yes, the equal area weighting has been taken into account.

*11. "line 270-273: The reader gets the impression, that only the "false positive" cases"were used here, but lines 268-269 state otherwise (all AIRS clear-sky, i.e. "false and correct positive"). Please clarify."*

Our statement is correct. Lines 268-269 in the original manuscript are for all AIRS clear-sky observations. We stated that the plot for them is not shown in the manuscript. Lines 270-273 are for "false positive" results only. The plot for them was shown in Figure 5 in the original manuscript (new Figure 6 in the revised manuscript).

Again, we are sincerely thankful to the reviewer for the thoughtful comments which have greatly improved the quality and clarity of our manuscript. We have acknowledged the reviewer in the text for the efforts.

Xiuhong Chen and Xianglei Huang